# Determination of Pharmaceuticals, Heavy Metals, and Oxysterols in Fish Muscle

**DOI:** 10.3390/molecules26051229

**Published:** 2021-02-25

**Authors:** Barbara Bobrowska-Korczak, Agnieszka Stawarska, Arkadiusz Szterk, Karol Ofiara, Małgorzata Czerwonka, Joanna Giebułtowicz

**Affiliations:** 1Department of Bromatology, Faculty of Pharmacy, Medical University of Warsaw, Banacha 1, 02-097 Warsaw, Poland; agnieszka.stawarska@wum.edu.pl (A.S.); malgorzata.czerwonka@wum.edu.pl (M.C.); 2Departments of Spectrometric Methods, National Medicines Institute, Chełmska 30/34, 00-725 Warsaw, Poland; a.szterk@nil.gov.pl (A.S.); k.ofiara@nil.gov.pl (K.O.); 3Department of Drug Analysis, Faculty of Pharmacy, Medical University of Warsaw, Banacha 1, 02-097 Warsaw, Poland; jgiebultowicz@wum.edu.pl

**Keywords:** pharmaceuticals, toxic elements, oxysterol, fish

## Abstract

The present study aimed to assess the levels of 98 multi-class pharmaceuticals including cardiovascular drugs, antidepressants, hypnotics, antibiotics, and sulfonamides occurring in the muscle tissue of fish caught in the Baltic Sea. The following fish species were collected: perch (*Perca fluviatilis*); flounder (*Platichthys flesus*); turbot (*Scophthalmus maximus*); plaice (*Pleuronectes platessa*); cod (*Gadus morhua callarias*); bream (*Abramis brama*); crucian (*Carassius carassius*). Additionally, in the examined fish muscle the levels of heavy metals and trace elements were determined (As; Ag; Au; Ba; Cd; Co; Cr; Cu; Hg; Li; Mo; Ni; Pb; Sb; Se; Sn; Tl; V) as well as the levels of cholesterol and its 5 derivatives (7-ketocholesterol; 7α-hydroxycholesterol; 7β-hydroxycholesterol; 5β,6β-epoxy-cholesterol; 5α,6α-epoxycholesterol). In the performed studies 11 out of 98 examined pharmaceuticals were detected in fish muscle. The levels of pharmaceuticals in fish muscle varied depending on the species. In the tissues of bream and crucian, no pharmaceuticals were found. Mercury, lead and arsenic were detected in the muscles of all examined fish. Based on the hazard factor for Hg, Pb, Cd, Ni (target hazard quotient, THQ < 1), it was found that the consumption of the studied fish does not constitute a health risk. However, the THQ for As remained >1 indicated possible risk from those metals. In the examined fish muscle the total cholesterol oxidation products (COPs) level of oxysterols were, respectively: 6.90 (cod) μg/g–4.18 μg/g (perch), which corresponded to 0.7–1.5% of cholesterol. The main COPs evaluated were 7-ketocholesterol (0.78 ± 0.14–1.79 ± 0.06 μg/g), 7β-hydroxycholesterol (0.50 ± 0.04–3.20 ± 2.95 μg/g) and 5β,6β-epoxycholesterol (0.66 ± 0.03–1.53 ± 0.66 μg/g). The assessment of health hazards due to contaminations is necessary, which may help to introduce national legislation and global standards aimed at reducing or even eliminating the exposure to contaminants.

## 1. Introduction

Fish is regarded as an excellent source of high-quality protein, particularly the essential amino acids lysine and methionine [1,2,3]. Globally, they comprise about 6 percent of dietary protein, yet it should be remembered that for some 3 billion people fish make up as much as 20 percent of the average per-capita intake of animal protein [4]. It is also known that fish and fish oils are a good source for the long-chain polyunsaturated fatty acids (PUFAs) especially eicosapentaenoic acid (EPA) and docosahexaenoic acid (DHA). Diet rich in fish lowers the risk of cardiovascular disease (CVD) [5,6,7]. Replacement of other dietary animal products with fish in the diet may be cardioprotective due to the reduced saturated fatty acid and cholesterol content of these products as well as their high polyunsaturated fatty acid content as compared with other animal protein sources. Omega-3 fatty acids are critical for neurological development in adults and an improvement in cognitive development in infants and young children. Fish is also high in dietary nutrients, such as calcium, selenium, and zinc [8,9]. In addition, all fish are a good source of B vitamins and, in the case of the fatty species of vitamins A and D [7].

Given the above-mentioned health benefits of fish, the 2010 Dietary Guidelines for Americans (DGA) recommend that people consume 8 ounces (227 g) of seafood per week, in particular, fatty saltwater fish such as salmon, mackerel, sardines, pompano, anchovies, swordfish, trout, and tuna to provide an average daily consumption of 250 mg of EPA/DHA [10]. National dietary guidelines for fish consumption in different countries recommended fish consumption in the amount between 100 g/week (Germany) to 550 g/week (Greece) [10,11,12,13].

Recent studies have shown that the awareness of the health benefits of fish consumption along with price and freshness of fish are the main factors affecting consumer decisions to purchase fish [14]. At the same time, it should be remembered that, despite multiple health benefits, fish can also contain various contaminants that are hazardous to health [15,16,17,18]. This brings about certain confusion as to the role of fish consumption in a healthy diet.

So far, the scientists have focused their attention mainly on such risks as the presence of methylmercury, dioxins, polychlorinated biphenyls, and parasites in fish [15,16,17,18,19,20,21]. However, there is a lack of data concerning the presence of certain pharmaceuticals in fish, in spite of a growing number of reports. In 2018 the total global pharmaceutical market was valued at about 1.2 trillion U.S. dollars [22,23]. This was a significant increase as compared with 2001 when the market was valued at just 390 billion U.S. dollars. In Poland, the pharmaceutical market has increased at a rate of 4.8% on average over the years 2016–2021 [22,23]. Due to significant drug usage by humans and animals, there are increasing amounts of pharmaceuticals in the environment. The main sources of environmental contamination in water by pharmaceuticals and their metabolites are sewages [24]. Pharmaceuticals and their metabolites enter wastewater treatment plants using wastewater from the disposal of unused or expired drugs in toilets, as well as by human excretion which is generally considered to be the primary source of pharmaceuticals in the environment [25]. The physicochemical properties of pharmaceuticals such as their durability, resistance to biodegradation, high solubility in water and low sorption coefficients contribute to a low level of their elimination during the water treatment processes for consumption purposes. Additionally, the drugs can derive from the pharmaceutical industry due to emission to the environment during production. Unfortunately, there is little or no understanding of the health risks from combined exposures to more than one contaminant and the means by which we can evaluate such interactions. The level of toxicity can vary depending on the route of exposure and dose, whereas personal characteristics such as the age and health condition may affect the individual’s response to toxic compounds. The contaminants can cause acute or chronic toxic effects. However, it should be noted that monitoring of the contaminants present in food and the investigations into the effect of pharmaceuticals contained in food on human health are very important [26].

Among metals mercury, cadmium, lead, and arsenic pose the greatest risk to human health [27,28,29]. They can contaminate both food and water and result in serious human health hazards, with the toxic effect. Lead toxicity affects almost all human organs, but the nervous system is most severely affected. In adults, prolonged lead exposure causes a reduction in cognitive development and intellectual performance. Lead contamination can also cause kidney damage, anemia and hypertension. Cadmium accumulates in the human body for a long time, causing damages to the nervous system, kidneys, bones, lungs, and cardiovascular system. It has been classified to be carcinogenic for humans (Group 1). Chronic exposure to arsenic increased risk of cardiovascular abnormalities, diabetes mellitus, neurotoxicity, hepatotoxicity and nephrotoxicity. Moreover, several studies showed that long-term exposure to arsenic induced carcinogenicity, especially cancer of the skin, bladder, and lungs. Chronic exposure to nickel has been linked to dermatotoxicity, lower body weight, and fetotoxicity in pregnant women. The main Hg, Cd, Pb, and As contamination factors in fluvial and marine sources are industrial and municipal wastewater discharge, mining, combustion of fossil fuels, deforestation, and fertilizers used in agriculture [29].

Oxysterols are 27-carbon derivatives of cholesterol created by enzymatic or radical oxidation, formed in the human body, or otherwise ingested in the diet [30,31,32]. The oxygenation of cholesterol occurs either on the side chain or on the sterol nucleus. Side-chain oxidation results in 24-hydroxycholesterol (24-HC), 25-hydroxycholesterol (25-HC), 27-hydroxycholesterol (27-HC) or other products whereas the oxidation of the steroid nucleus gives rise to ring oxysterols, mostly ring-B oxysterols, e.g., 6-hydroxycholesterol (6-HC), 7α/β-hydroxycholesterol (7α/β-HC), 7-ketocholesterol (7-KC). In very low concentrations, oxysterols are natural components of the human body. By modulating the activity of many proteins, e.g., liver X receptors, oxysterol-binding proteins, or some ATP binding cassette transporters, oxysterols can exert a notable influence on certain cellular functions and various physiological processes, for instance on cholesterol metabolism, membrane fluidity regulation, or intracellular signaling pathways. However, oxysterol action is also associated with human pathologies, e.g., cancer, atherosclerosis, diabetes mellitus type 2, and neurodegenerative disorders (Alzheimer’s disease, Parkinson’s disease, Huntington’s disease). Oxysterols can also induce malignancies, such as breast, prostate, colon, and bile duct cancer [30,31,32].

The aim of the present study was to assess the levels of 98 multi-class pharmaceuticals from different therapeutic classes including cardiovascular drugs, antidepressants, antibiotics, and sulfonamides in the muscle tissue of fish caught in the Baltic Sea. Additionally, in the examined fish muscle the levels of heavy metals and trace elements were determined (As; Ag; Au; Ba; Cd; Co; Cr; Cu; Hg; Li; Mo; Ni; Pb; Sb; Se; Sn; Tl; V) as well as the levels of cholesterol and its 5 derivatives (7-ketocholesterol; 7α-hydroxycholesterol; 7β-hydroxycholesterol; 5β,6β-epoxycholesterol; 5α,6α-epoxycholesterol). The following fish species were collected: perch (*Perca fluviatilis*) (*n* = 6); flounder (*Platichthys flesus*) (*n* = 7); turbot (*Scophthalmus maximus*) (*n* = 6); plaice (*Pleuronectes platessa*) (*n* = 7); cod (*Gadus morhua callarias*) (*n* = 6); bream (*Abramis brama*) (*n* = 6); crucian (*Carassius carassius*) (*n* = 6).

## 2. Results

### 2.1. Determination of Selected Pharmaceuticals

Real sample determination showed that 11 of the examined 98 compounds (Table 1) from the following therapeutical families were detected in fish muscle: antibiotics, sulfonamides, antidepressants, anticonvulsants, antipsychotics, antiparasitics and cardiovascular drugs. The highest concentration was observed for ofloxacin (up to 3.43 ng/g, cod) and thiabendazole (up to 2.09 ng/g, turbot) followed by metronidazole (max 1.92 ng/g, turbot), promazine (max 1.56 ng/g, cod), carbamazepine (max 1.18 ng/g, cod), fluoxetine (max 0.57 ng/g, perch), tianeptine (max 0.53 ng/g, perch), clarithromycin (max 0.44 ng/g, flounder) sulfadimethoxine (max 0.37, flounder), bisoprolol (max 0.23 ng/g, plaice) and erythromycin (max 0.17 ng/g, cod). The levels of pharmaceuticals in fish muscle varied depending on the species. In the tissues of bream and crucian no pharmaceuticals were found. The list of pharmaceuticals which were not detected in any of the examined fish species is presented in Table 2.

### 2.2. Determination of Selected Trace Elements

Summaries of chemical analysis of the contents of heavy metals and trace elements detected in the Baltic fish are illustrated in Table 3.

Mercury, lead and arsenic were detected in the muscles of all examined fish. The As content was the highest reaching 960 ± 190 μg/kg (flounder), followed by lead (up to 120 ± 120 μg/kg, perch) and mercury (up to 68 ± 26 μg/kg, perch). Cadmium was detected in flounder, turbot, plaice and cod with the highest concentration of 14.90 ± 0.96 μg/kg (plaice). The highest levels of mercury were found in perch and turbot, the highest content of arsenic was detected in flounder whereas the highest level of lead was found in perch. The level of nickel in the examined fish muscle varied in the range from 16.6 ± 8.4 μg/kg (bream) to 39.4 ± 5.6 μg/kg (perch).

The estimated daily intake (EDI) of the heavy metals was measured from the consumption of the seven species considering the mean fish consumption by the citizens of selected Baltic states (Table 4). Therefore, the EDI (μg/kg body weight/day) of Hg, Pb, Cd, As, Ni were between: 0.02 (Germany) and 0.04 (Denmark, EU) (μg/kg body weight/day); 0.03 (Estonia, Germany, Poland) and 0.05 (Denmark, EU) (μg/kg body weight/day); 0.006 (Germany, Poland) and 0.01 (Denmark, EU) (μg/kg body weight/day); 0.25 (Germany) and 0.45 (Denmark) (μg/kg body weight/day); 0.02 (Estonia, Germany, Poland, Russian Federation) and 0.03 (Denmark, EU) (μg/kg body weight/day), respectively. Additionally, the target hazard quotient (THQ) was calculated (Table 4). The THQ results for different population groups were found to be between: 0.08 μg/kgBW/day (Germany, Poland) and 0.14 μg/kgBW/day (Denmark) for Hg; 0.007 μg/kgBW/day (Germany, Poland) and 0.012 μg/kgBW/day (Denmark, EU) for Pb; 0.006 μg/kgBW/day (Germany, Poland) and 0.010 μg/kgBW/day (Denmark, EU) for Cd; 0.84 μg/kgBW/day (Germany) and 1.49 μg/kgBW/day (Denmark) for As; 0.001 μg/kgBW/day (Estonia, Germany, Poland, Russian Federation) and 0.002 μg/kgBW/day (Denmark, EU) for Ni. As contributed with the highest daily intake.

### 2.3. Determination of Oxysterols

The level of cholesterol in the examined fish muscle was between 283 ± 39 μg/g (plaice) and 600 ± 130 μg/g (perch) (Table 5). The total COPs was the highest for cod and the lowest for perch at the level which corresponded to 0.7–1.5% of cholesterol. The highest contribution in COPs had 7-ketocholesterol, 7β-hydroxycholesterol and 5β,6β-epoxycholesterol.

## 3. Discussion

The determinations of the contents of pharmaceuticals, heavy metals and trace elements as well as the levels of cholesterol and its five derivatives allowed us to evaluate the level of environmental contamination and fish consumption safety of the fish caught in Polish inshore waters of the Baltic Sea. This study aimed to determine the concentration of 98 pharmaceuticals in fish muscle. We observed the occurrence of 11 out 98 examined from the following therapeutic groups: antibiotics, sulfonamides, antidepressants, anticonvulsants, antiparasitics, antipsychotics and cardiovascular drugs. The pharmaceuticals were previously detected in Polish surface waters as well [37,38,39,40]. The highest concentration in fish muscles (>2.0 ng/g) was observed for ofloxacin and thiabendazole. Thiabendazole is a fungicide and parasiticide. Its concentration in Polish surface waters was described to be from <0.7–16 ng/L and its removal in wastewater treatment plant (WWTP) was negative, that means that the concentration in effluent was higher than in influent [37,40]. The steady state bioconcentration factor (BCF) of thiabendazole for edible tissues of bluegill sunfish was 22.84 L/kg, so no bioconcentration in fish tissues is expected [41]. Thus, the reason of high concentration of the compound in fish muscles should be explored further towards e.g., higher concentrations of thiabendazole in the place of fish existence, higher BCF in the analyzed fish species or importance of food exposure pathway in the compound accumulation [42]. The second pharmaceutical detected at the highest concentration, ofloxacin, was also detected in Polish surface waters at relatively low concentrations i.e., 8.4–31 ng/L [37,40] and did not tend to accumulate in carp (*Cyprinus carpio)* muscles (BCF = 0.04 L/kg) [43]. Thus, similarly to thiabendazole the measured concentration is higher than expected and the reason should be further explored.

Pharmaceuticals are detected in fish tissues not only in Poland, but also worldwide [44]. Arya et al. [45] detected hormones, anti-inflammatory drugs (naproxen, diclofenac), fibrate (gemfibrozil) and anticolvulsant (carbamazepine) in 17–33% of the analyzed fish samples from VA, USA. Trout (*Salmo truta)* bought in Spain reached the highest concentrations of bisphenol A, metoprolol, propranolol, ibuprofen, sulfamethoxazole and azithromycin, all above 10 ng/g [46]. In another study from the USA [47], the occurrence of pharmaceuticals in eight wild fish species collected from 26 sampling sites downstream from WWTPs (USA) was analyzed. Thirteen pharmaceuticals (out of 20 compounds analyzed) were quantified in fish fillets at concentrations mostly below 10 ng/g. The psychoactive drugs venlafaxine, carbamazepine, and its metabolite 2-hydroxycarbamazepine were the most prevalent compounds (58%, 27%, and 42%, respectively). Carbamazepine was detected at concentrations up to 8 ng/g, together with its metabolite 2-hydroxy-carbamazepine, which was found at lower concentrations (up to 2.5 ng/g) (detection frequencies were 16% and 20%, respectively). The drugs from this group are frequently prescribed to patients by doctors and their consumption has considerably increased in western countries over the last decade and was also detected in fish muscles in our study. Carbamazepine, primarily used for the treatment of epilepsy, is excreted by humans in its unchanged form along with several metabolites, including the metabolite 2-OH-CBZ. The CBZ:2-OH-CBZ ratio obtained was found to range from 14:1 to 6:1 for all the organs of Onesided livebearer *(Jenynsia multidentate*). However, the metabolization of carbamazepine seems to depend on species, since 2-OH-CBZ was not detected in biota samples from the carbamazepine exposure studies carried out on mussels (*Mytilus galloprovincialis*) or zebra mussel (*Dreissena polymorpha)*, nor in in vitro studies using rainbow trout liver fractions [35,48,49]. Salbutamol, a drug used to treat asthma, and the diuretic hydrochlorothiazide were also frequently detected (in over 20% of the samples).

From the performed studies it results that the level of pharmaceuticals detected in fish muscle varies depending on fish species. In the fish muscle of such species as bream or crucian, none of the examined drugs were found. Bioaccumulation of pharmaceuticals also depends on such factors as the differences in lipid concentration of the examined fish (those with a higher lipid level have a higher ability to accumulate hydrophobic compounds), size (larger-bodied animals have slower elimination rates), or life stage. The accumulation also depends on pharmaceutical concentration, pH of the environment, log D of the pharmaceuticals and simultaneous exposure to other compounds. Moreover, it should also be highlighted that not only direct exposure, but also exposure pathway through food webs could be important for some pharmaceuticals because these compounds could also bioaccumulate in lower levels of food chains [42]. Thus, so far from now the concentrations of pharmaceuticals in fish muscles are hard to predict and should be determined.

Multiple studies have confirmed the presence of drugs and their derivatives both in the environment and in food, including fish. Therefore, it is necessary to perform constant monitoring of food contents as well as to carry out investigations of health risk to consumers in order to assess food safety. The determination of the concentration of pharmaceuticals in particular elements of the environment as well as in food should be treated as a priority task as concerns monitoring the environment and food control. The presence of pharmaceuticals in the environment and food is also a new challenge for the technologies of water purification and sewage treatment, where new techniques must be developed that would effectively eliminate the discussed compounds in case the health risk would be confirmed.

In the presented research mercury, lead and arsenic were detected in the muscles of all examined fish. In opposite to flounder, turbot, plaice and cod in the tissues of perch, bream, and crucian no cadmium was found. None of the examined samples exceeded the maximum permissible levels established for mercury, cadmium, and lead by the respective directives of the European Union Commission (respectively: EU Commission Directive No. 420/2011 of 29 April 2011 (0.50 mg/kg wet weight Hg); EU Commission Directive No. 488/2014 of 12 May 2014 (0.050 mg/kg wet weight Cd); EU Commission Directive No. 2015/1005 of 25 June 2015 0.3 mg/kg wet weight Pb) and other international organizations [50,51,52,53]. The observations are similar those found by other authors. Findings reported by Łuczyńska et al. [54] showed that flounder (0.056 mg/kg) had more mercury than herring (0.021 mg/kg), bream (0.016 mg/kg), rainbow trout (0.015 mg/kg) and carp (0.006 mg/kg) (*p* ≤ 0.05). Positive correlation coefficients were found between mercury levels in the muscles and the fish weight and length. However, a significant correlation was found between body weight and the content of mercury in muscle tissue of flounder (*p* = 0.008) and herring (*p* = 0.0005). According to the results obtained by Ullah et al. [55] the concentrations of heavy metals in the muscles of ilish (*Tenualosa ilisha)* and shad mud (*Dorosoma cepedianum)* fish species were measured as 0.725–1.631 mg/kg for Pb, 0.020–1.092 mg/kg for Cd, 0.374–1.001 mg/kg for As and <0.02 mg/kg for Hg. The ranking order of mean concentration of heavy metals was found to be Pb (1.342 mg/kg) > As (0.715 mg/kg) > Cd (0.462 mg/kg) > Hg (<0.002 mg/kg) [55].

The exceedance of the recommended values of toxic metals set by different regulatory bodies does not always represent the human health risk, and consequently, in recent years, the health risk assessment has been extensively used to evaluate the impact of the hazards of heavy metals bioaccumulation on human health. Assessment of potential risk of heavy metals on human health is evaluated by estimated daily intake, target hazard quotient, and other indexes based on the fish consumption [3,55]. Based on the hazard factor for Hg, Pb, Cd, Ni (THQ < 1), it was found that the consumption of the studied fish does not constitute a carcinogenic health effects. Their consumption does not raise concerns related to adverse health risks. However, the THQ for As remained >1 indicating possible risk from this metal. About 90% of human exposure to As is due to the intake consumption of fish, shellfish, and/or other marine organisms. It should be considered in future evaluations of environmental contaminants along with their speciation (organic and inorganic As), especially in different aquatic organisms of commercial interest. The toxicity of arsenic compounds has been reported to follow the order; As (III) > As (V) > monomethylarsonate (MMA) > dimethylarsinic acid (DMA) > organic arsenic species. The inorganic arsenic species As (III) and As (V) have been classified by the International Agency for Research on Cancer as Class I chemicals (carcinogenic to humans), meanwhile, MMA and DMA species are classified as Class II B chemicals (possibly carcinogenic to humans). Health effects from arsenic are generally associated with exposure to As (III) and As (V). Inorganic arsenic usually makes up less than 10% of the total arsenic in fish muscle. Exposure to elevated levels of arsenic has been reported in areas worldwide, including Bangladesh, Taiwan, India, Mexico, China, Chile, Argentina, and South America [27,28].

In the examined fish muscle the total cholesterol oxidation products level corresponded to 0.7–1.5% of cholesterol. Standing out among the main COPs evaluated were 7-ketocholesterol, 7β-hydroxycholesterol and 5β,6β-epoxycholesterol. Similar results were obtained by other authors. Pickova and Dutta [56] assessed the presence of COPs in fresh fish roe: salmon (*Salmo salar*), lumpsucker (*Cyclopterus lumpus*), and also black-colored lumpsucker roe (*Cyclopterus lumpus*). The authors reported the presence of the following COPs: 7α-hydroxycholesterol, 7β-hydroxycholesterol, 5,6β-epoxycholesterol 5,6α-epoxycholesterol, cholestanetriol, 25-hydroxycholesterol, 20α-hydroxycholesterol, and 7-ketocholesterol. The total concentration of the COPs was between 6.23 and 93.06 μg/g in all examined samples. Oliveira et al. [57] determined the levels of cholesterol oxides in mandim fish (*Arius spixii*) in the fresh, salted, and dried forms. According to the authors, 7-ketocholesterol was the only oxide detected in the study, and in the fresh mandim fish the levels were 8.31 μg/g, although in dried samples the levels reached 17.90 μg/g (*p* < 0.05). The formation and levels of cholesterol oxides in seabob shrimp (*Xiphopenaeus kroyeri*, Heller, 1862) were examined by Lira et al. [58]. In the fresh samples of the COPs 7-ketocholesterol, 7α-hydroxycholesterol, and 7β-hydroxycholesterol were found in varying concentrations (4.3 to 38.7 μg/g, dry mass). However, after smoking the authors observed cholesterol losses of 27%, and 7β- hydroxycholesterol losses of 50%, whereas 7-ketocholesterol and 7α-hydroxycholesterol levels remained unchanged.

Kao and Hwang [59] analyzed COPs in dehydrated squid and assessed the presence of the following COPs: 7α-hydroxycholesterol, 7β-hydroxycholesterol, 5,6α-epoxycholesterol, 5,6β-epoxycholesterol, 7-ketocholesterol, 20α-hydroxycholesterol, 25-hydroxycholesterol, and cholestanetriol. The authors reported that after heating at 200 °C for 10 min, the total COPs level increased from 12.07 to 43.46 μg/g.

Oxidized cholesterol can be about 100 times more toxic than regular cholesterol thus raising additional concern about the healthy aspects of eating certain foods such as fish. The different processing techniques (including salting, smoking, heating, canning, freezing) occur to be key factors in the cholesterol oxidation process. However, as it was shown in our studies, oxysterols can be found even in raw fish. The aim of investigations on the concentration of oxysterols in food is to focus general attention on the problem of their presence and possible bad effects on human health. Also, the public awareness of the presence of oxysterols in fish might help to introduce new methods of their elimination.

## 4. Materials and Methods

### 4.1. Sampling

This paper reports the levels of pharmaceuticals, heavy metals and trace elements, including toxic ones: Hg, Pb, Cd, and Ni, as well as oxysterols in fish that were caught from the Polish Baltic fishing areas. The samples of fish to be examined were purchased directly from local fishermen in Gdańsk, Łeba, and Stegna fishing areas immediately upon landing. The following fish species were collected: perch (*Perca fluviatilis*) (*n* = 6); flounder (*Platichthys flesus*) (*n* = 7); turbot (*Scophthalmus maximus*) (*n* = 6); plaice (*Pleuronectes platessa*) (*n* = 7); cod (*Gadus morhua callarias*) (*n* = 6); bream (*Abramis brama*) (*n* = 6); crucian (*Carassius carassius*) (*n* = 6). The samples were collected and stored at −70 °C.

### 4.2. Determination of Selected Drug Substances

All the pharmaceutical standards were purchased from Sigma–Aldrich (Hamburg, Germany) or were a gift from the Drug Research Institute (Warsaw, Poland), and were of high purity grade (>98%). Acetaminophen D4, caffeine D9, salbutamol D7, trimethoprim D9, fluoxetine D5, bisoprolol D5, ciprofloxacin D8, mycophenolic acid D3, clindamycin D3, valsartan D3, erythromycin C13D3, dulfamethoxazole D4 were used as internal standards and were purchased from Toronto Research Chemicals (Toronto, Ontario, ON, Canada). The solvents, HPLC gradient grade methanol, acetonitrile (LiChrosolv), and formic acid 98%, were provided by Merck (Hamburg, Germany). Ultrapure water was obtained from a Millipore water purification system (Milli-Q water).

Tissue samples (50 mg) were homogenized in 1.5 mL of acetonitrile with 0.1% formic acid using a manual glass homogenizer. The internal standard was added to the final concentration of 10 ng/mL. The sample was frozen at −20 °C (10 min) and centrifuged at 10,000 RPM (5 min) (Centrifuge MiniSpin^®^ Plus, Eppendorf, Hamburg, Germany). To the supernatant 300 mg of ammonium acetate was added. The sample was stirred for 1 min and centrifuged at 10,000 RPM for 5 min. Afterwards, the upper layer of the extract was placed in a tube with C_18_ sorbent (100 mg), vortexed (1 min), and centrifuged at 10,000 RPM for 5 min again. The extract was evaporated to dryness under nitrogen at 40 °C. The dry residue was dissolved in 100 µL of the mobile phase, centrifuged at 10,000 g for 5 min, and placed into the microvial.

Liquid chromatography analysis was performed on Agilent 1260 Infinity (Agilent Technologies, Santa Clara, CA, USA) connected in series to a 4000 QTRAP mass spectrometer (AB Sciex, Framingham, MA, USA) equipped with a Turbo Ion Spray source that was operated in a positive mode. The details can be found in Table 6. The target compounds were analyzed in MRM mode. Limit of detection (LOD) and limit of quantitation (LOQ) for the entire method (including extraction) was determined as the amount of analyte with a signal-to-noise ratio (S/N) of 3:1 and 10:1, respectively. The LOD and LOQ were performed on a blank matrix. The matrix-matched calibration was used to quantify the analyses. The method precision for all detected compounds was lower than 15%.

### 4.3. Determination of Heavy Metals and Trace Elements

All solvents and reagents were of the highest commercially available purity grade. Ultrapure water (resistivity 18 MΩ/cm) was obtained from a NANOPURE DIAMON UV system (Barnstead, NH, US) and was used for the preparation of all standards and sample solutions. Suprapur grade 65% HNO_3_ and 37% HCl (Merck) were used for sample dissolutions. The multielement solution at a concentration of 10 mg/L each Ag, As, Au, Ba, Cd, Co, Cr, Cu, Hg, Ir, Li, Mo, Ni, Os, Pb, Pd, Pt, Rh, Ru, Sb, Se, Sn, Tl, V was purchased from INORGANIC Ventures (Christiansburg, VA, USA). The purity of plasma gas (argon) and cell gas (helium) was greater than 99.999%.

One g of each fish sample was weighted and inserted directly into a microwave-closed vessel. One mL of HCl and 4 mL of HNO_3_ was added. The digestion of the fish samples was performed using a high-pressure laboratory microwave oven (UltraWAVE T640, Milestone, Shelton, CT US). The heating program was performed in two steps. In the first step, the temperature was increased linearly from 25 to 210 °C in 15 min. In the second step, the temperature was held at 210 °C for 8 min. After the digestion procedure, the samples were diluted to the final volume of 100 mL with water.

A quadrupole ICP-MS 7800 (Agilent Technologies, Tokyo, Japan) equipped with an octopole collision cell was used for all analyzed heavy metals and trace elements. Internal Standard was added to compensate for any effects from acids or instrument drift. The measurements were made with a nickel sampler and skimmer cones. The ICP-MS operational conditions are summarized in Table 7.

### 4.4. Determination of Oxysterols

Cholesterol, oxysterols, squalene analytical standards, silylation mixture and internal standard were purchased from Merck. All solvents were HPLC gradient grade.

To 300 mg of tissue, 3 mL of 1 M KOH in ethanol, 25 μL of internal standard solution (5α-cholestane in hexane, 0.5 mg/mL) and 10 μL solution (butylated hydroxytoluene in ethanol, 5 mg/mL) were added, homogenized and left for 20 h. Then 4 mL of water, 2 mL of hexane were added and shaken. The hexane layer was transferred to a 2 mL vial and evaporated to dryness under a stream of nitrogen. 50 μL of pyridine and 25 μL silylation mixture (N,O-bis(trimethylsilyl)trifluoroacetamide with 1% of trimethylchlorosilane, for GC derivatization were added to the dry residue and mixed thoroughly. The derivatization process was carried out at a temperature of 80 °C for 40 min. After silylation, 225 μL of hexane was added and placed into the microvial.

Gas chromatography with time-of-flight mass spectrometry (Pegasus^®^ BT, LECO Corporation, St. Joseph, MO, USA) was used to determine cholesterol and its oxidized derivatives (COPs). Parameters of analysis are presented in Table 8. Cholesterol, squalene, and COPs trimethylsilyl derivatives were identified based on retention times and mass spectra. The target compounds were analyzed in TIC mode. Quantitative analysis was based on standard curves was made for each compound and internal standard. LOD and LOQ were determined as the amount of analyte with a signal-to-noise ratio (S/N) of 3:1 and 10:1, respectively.

### 4.5. Health Risk Assessment for Fish Consumption

#### 4.5.1. Estimated Daily Intake of Heavy Metals (EDI)

The estimated daily intake (EDI) (μg/kg body weight/day) of heavy metals (Hg, Pb, Cd, As, Ni) was calculated using the following equation:(1)$$\mathrm{EDI}=\frac{C\cdot \mathrm{IR}}{\mathrm{BW}}$$
where: C is the average concentration of heavy metal in the fish muscle tissue (μg/g); IR is the daily ingestion rate (g/day); BW is the average body weight (60 kg) [54,60].

#### 4.5.2. Target Hazard Quotients (THQ)

The THQ estimated the non-carcinogenic health risk of consumers due to the intake of heavy metal contaminated fish use an oral reference dose (RfD) of Hg, Pb, Cd, As, Ni [53,59]. The reference doses (mg/kgBW/day) defined as the maximum tolerable daily intake of a specific metal that does not result in any deleterious health effects (Hg = 0.0003; Pb = 0.004; Cd = 0.001; As = 0.0003; Ni = 0.02 mg/kg/day) [3,61,62]. When THQ < 1, there is a health benefit from fish consumption and the consumers are safe, whereas THQ > 1 suggests a high probability of adverse effects human health. THQ was calculated using the following equation:(2)$$\mathrm{THQ}=\frac{\mathrm{EFr}\cdot \mathrm{ED}\cdot \mathrm{FiR}\cdot C}{\mathrm{RfD}\cdot \mathrm{BW}\cdot \mathrm{TA}}\;\times \;10^{-3}$$
where: EFr is the exposure frequency (365 days/year); ED is the exposure duration (70 years); FiR is the fish ingestion rate (g/person/day); C is the average concentration of heavy metal in the fish muscle tissue (μg/g); RfD is the oral reference dose (mg/kg/day); BW is the average body weight; TA is the average exposure time (365 day/year × ED) [54,60].

### 4.6. Statistical Analysis

All data were presented as mean values ± standard deviation. Due to lack of normal distribution of variables in groups, as assessed by Shapiro–Wilk test, data were analyzed by a Kruskal–Wallis test (*p* = 0.05) followed by a post hoc test for multiple comparisons of mean rank (*p* = 0.05); letters indicated homogeneous groups. The data were evaluated using STATISTICA 13.3 software (StatSoft, Krakow, Poland).

## 5. Conclusions

The results of the present study provided new information on the concentrations of pharmaceuticals, heavy metals, and oxysterols in the muscle tissue of fish caught in the Baltic Sea. In the performed studies 11 out of 98 examined pharmaceuticals (including antibiotics, sulfonamides, antidepressants, and cardiovascular drugs) were detected in fish muscle. The health risk analysis indicated safe levels for individual heavy metals except for as where the target hazard quotient (THQ) exceeded 1. Also, the public awareness of the presence of oxysterols in fish might help to introduce new methods of their elimination. Contamination in the aquatic ecosystem is considered as serious problem at the global level due to the adverse effects on human health. Monitoring plays a vital role in food safety and may help to introduce national legislation and global standards aimed at reducing or even eliminating exposure to contaminants.

## Figures and Tables

**Table 1 molecules-26-01229-t001:** The maximum concentration of pharmaceuticals detected in fish muscles (ng/g) and the number of positive findings (n_p_).

Pharmaceuticals	Perch (*n* = 6)	Flounder (*n* = 7)	Turbot (*n* = 6)	Plaice (*n* = 7)	Cod (*n* = 6)	Bream (*n* = 6)	Crucian(*n* = 6)	LOD *
Bisoprolol	0.21 (n_p_ = 1)	<0.06 (n_p_ = 1)	<LOD	0.23 (n_p_ = 2)	0.18 (n_p_ = 1)	<LOD	<LOD	0.02
Carbamazepine	<LOD	1.12 (n_p_ = 1)	0.62 (n_p_ = 1)	<LOD	1.18 (n_p_ = 6)	<LOD	<LOD	0.04
Clarithromycin	0.11 (n_p_ = 1)	0.44 (n_p_ = 1)	0.20 (n_p_ = 1)	0.13 (n_p_ = 1)	0.42 (n_p_ = 3)	<LOD	<LOD	0.02
Erythromycin	<LOD	0.04 (n_p_ = 3)	<0.03 (n_p_ = 2)	0.03 (n_p_ = 2)	0.17 (n_p_ = 2)	<LOD	<LOD	0.01
Fluoxetine	0.57 (n_p_ = 1)	<LOD	<LOD	<LOD	<LOD	<LOD	<LOD	0.07
Metronidazole	0.33 (n_p_ = 3)	0.36 (n_p_ = 2)	1.92 (n_p_ = 1)	0.28 (n_p_ = 3)	0.47 (n_p_ = 3)	<LOD	<LOD	0.03
Ofloxacin	<LOD	<LOD	0.58 (n_p_ = 2)	2.64 (n_p_ = 2)	3.43 (n_p_ = 2)	<LOD	<LOD	0.01
Promazine	<LOD	0.04 (n_p_ = 1)	<LOD	<LOD	1.56 (n_p_ = 1)	<LOD	<LOD	0.02
Sulfadimethoxine	<LOD	0.37 (n_p_ = 1)	<LOD	<LOD	<LOD	<LOD	<LOD	0.10
Thiabendazole	0.52 (n_p_ = 1)	<LOD	2.09 (n_p_ = 1)	<0.03	0.74 (n_p_ = 1)	<LOD	<LOD	0.03
Tianeptine	0.53 (n_p_ = 2)	0.13 (n_p_ = 1)	0.23 (n_p_ = 2)	<LOD	0.27 (n_p_ = 1)	<LOD	<LOD	0.05

* LOD: limit of detection (ng/g).

**Table 2 molecules-26-01229-t002:** Pharmaceuticals and selected metabolites which were not detected in fish muscle with their limit of detection (ng/g).

Pharmaceutical	LOD *	Pharmaceutical	LOD	Pharmaceutical	LOD	Pharmaceutical	LOD	Pharmaceutical	LOD
1-Naphthoxyacetic acid	0.88	Doxepin	0.10	Mebendazole	0.16	Paroxetine	0.07	Sulfanilamide	0.22
Acebutolol	0.09	Drotaverine	0.10	Metformin	0.17	Pefloxacin	0.06	Sulfathiazole	0.03
Amitriptyline	0.16	Duloxetine	0.12	Methoxyverapamil	0.07	Piperacillin	0.23	Telmisartan	0.16
Amlodipine	0.08	Enalapril	0.39	Metoprolol	0.06	Propafenone	0.02	Tetracycline	0.83
Atenolol	0.16	Escitalopram	0.04	Mianserin	0.15	Propranolol	0.09	Tiamulin	0.23
Azithromycin	0.10	Fenofibrate	0.16	Mirtazapine	0.13	Protriptyline	0.04	Tianeptine	0.05
Bosentan	0.07	Fleroxacin	0.12	Moclobemide	0.08	Pseudoephedrine	0.09	Tolperisone	0.67
Cefotaxime	0.61	Fluconazole	0.12	Morantel	0.12	Quinapril	0.1	Trazodone	0.05
Cefotaxime	0.61	Fluvoxamine	0.12	Mycophenolic acid	0.22	Ramipril	0.19	Trimethoprim	0.03
Chlorpromazine	0.14	Guaifenesin	0.18	Nalidixic acid	0.18	Ranitidine	0.03	Trimipramine	0.16
Chlortetracycline	0.69	Imipramine	0.08	Nifedipine	0.42	Roxithromycin	0.29	Tylosin	0.07
Clindamycin	0.07	Labetalol	0.31	Norfloxacin	0.41	Salbutamol	0.12	Valsartan	0.06
Clomipramine	0.04	Levofloxacin	0.51	Nortriptyline	0.07	Sertraline	0.11	Venlafaxine	0.05
Codeine	0.16	Lincomycin	0.23	Omeprazole	0.06	Sotalol	0.03	Verapamil	0.21
Desipramine	0.10	Lomefloxacin	0.21	Opipramol	0.05	Sulfadiazine	0.17	Xylometazoline	0.03
Dextromethorphan	0.08	Losartan	0.05	Oxymetazoline	0.16	Sulfamethazine	0.05		
Diclofenac	0.18	Lovastatin	0.21	Oxytetracycline	0.41	Sulfamethoxazole	0.06		
Diltiazem	0.14	Maprotiline	0.04	Pantoprazole	0.08	Sulfamethoxazole	0.12		

* LOD: limit of detection (ng/g).

**Table 3 molecules-26-01229-t003:** The levels of heavy metals and trace elements in the fish muscle of fish species (mean ± SD) (μg/kg).

Heavy Metals and Trace Elements	Perch	Flounder	Turbot	Plaice	Cod	Bream	Crucian
As (arsenic)	350 ± 270 ^a,b^	960 ± 190 ^c^	470 ± 42 ^a^	454 ± 66 ^a^	355 ± 10 ^a,b^	50.0 ± 2.3 ^b^	108.7 ± 1.3 ^a,b^
Ag (silver)	11.2 ± 3.1 ^a,b^	56 ± 43 ^a^	14.6 ± 5.3 ^a^	19.6 ± 3.1 ^a^	6.91 ± 0.51 ^b^	9.75 ± 0.88 ^a,b^	<LOD *
Au (gold)	19.5 ± 1.5 ^c^	78.7 ± 4.7 ^a,b^	40.5 ± 4.0 ^b,c^	78 ± 19 ^a^	38.0 ± 2.4 ^c^	43 ± 37 ^c^	17.21 ± 0.98 ^c^
Ba (barium)	50.1 ± 3.4 ^b,c^	150 ± 110 ^b^	94.4 ± 7.1 ^b^	311 ± 45 ^a^	26.9 ± 7.8 ^c^	390 ± 130 ^a^	213 ± 26 ^a,b^
Cd (cadmium)	<LOD	1.02 ± 0.84 ^c^	11.19 ± 0.93 ^a,b^	14.90 ± 0.96 ^a^	9.18 ± 0.21 ^b,c^	<LOD	<LOD
Co (cobalt)	<LOD	7.4 ± 2.6 ^b,c^	26.4 ± 2.6 ^a^	11 ± 13 ^b,c^	21.63 ± 0.53 ^a,b^	1.30 ± 0.11 ^c^	<LOD
Cr (chromium)	78 ± 14 ^a,b^	46.6 ± 9.2 ^b,c^	37.8 ± 6.9 ^c,d^	33.0 ± 6.9 ^c,d^	20.5 ± 0.71 ^d^	88 ± 17 ^a^	227 ± 71 ^a^
Cu (copper)	477 ± 62 ^a,b^	318 ± 78 ^c,d^	423 ± 72 ^a,b^	235 ± 22 ^d,e^	121.6 ± 3.9 ^e^	350 ± 130 ^b–d^	857 ± 63 ^a^
Hg (mercury)	68 ± 26 ^a,b^	39.3 ± 5.6 ^b^	66.6 ± 3.7 ^a^	23 ± 12 ^c,d^	36.8 ± 2.2 ^b,c^	21 ± 18 ^c,d^	6.55 ± 0.75 ^d^
Li (lithium)	<LOD	6.4 ± 4.0 ^a^	13.4 ± 3.0 ^b^	7.0 ± 2.5 ^a^	<LOD	2.8 ± 1.3 ^a^	<LOD
Mo (molybdenum)	<LOD	0.56 ± 0.24 ^c^	11.43 ± 0.64 ^a^	10.89 ± 0.49 ^a,b^	8.07 ± 0.27 ^b,c^	<LOD	<LOD
Ni (nickel)	39.4 ± 5.6 ^a^	22.6 ± 2.2 ^b,c^	25.8 ± 4.5 ^a,b^	28 ± 19 ^b,c^	20.5 ± 2.0 ^b,c^	16.6 ± 8.4 ^c^	31.1 ± 8.6 ^a–c^
Pb (lead)	120 ± 120 ^a^	31.3 ± 3.6 ^a^	31.5 ± 9.5 ^a^	38 ± 19 ^a^	12.8 ± 2.3 ^b^	21.7 ± 4.7 ^a,b^	31.8 ± 8.2 ^a^
Sb (antimony)	<LOD	<LOD	12.03 ± 0.83 ^a^	13.7 ± 1.1 ^a^	10.66 ± 0.26 ^b^	<LOD	<LOD
Se (selenium)	376 ± 12 ^a^	277 ± 21 ^b,c^	319 ± 48 ^a,b^	209 ± 15 ^d^	286 ± 13 ^a–c^	253 ± 56 ^c,d^	193 ± 6.9 ^d^
Sn (tin)	<LOD	42.4 ± 5.5 ^a^	54 ± 44 ^a^	33 ± 21 ^a^	159.3 ± 5.8	32.7 ± 2.9 ^a^	<LOD
Tl (thallium)	<LOD	10.30 ± 0.67 ^c^	20.7 ± 1.9 ^a^	15.4 ± 7.0 ^a–c^	19.8 ± 0.54 ^a,b^	10.8 ± 1.1 ^b,c^	<LOD
V (vanadium)	<LOD	0.27 ± 0.16 ^b^	8.02 ± 0.59 ^a^	13.87 ± 0.58 ^a^	6.58 ± 0.18 ^b^	<LOD	<LOD

^a–e^ homogenous groups in rows (multiple comparisons of mean rank; *p* < 0.050); * LOD: limit of detection.

**Table 4 molecules-26-01229-t004:** Health risk assessment of heavy metals for fish consumption (μg/kgBW/day) [33,34,35,36].

Country	Per Capita Fish Consumption(g Capita/Day)	Heavy Metal									
		Hg		Pb		Cd		As		Ni	
		EDI ^1^	THQ ^2^	EDI	THQ	EDI	THQ	EDI	THQ	EDI	THQ
Denmark	68.6	0.04	0.14	0.05	0.012	0.010	0.010	0.45	1.49	0.03	0.002
Estonia	44.3	0.03	0.09	0.03	0.008	0.007	0.007	0.29	0.97	0.02	0.001
EU	67.1	0.04	0.14	0.05	0.012	0.010	0.010	0.44	1.46	0.03	0.002
Germany	38.6	0.02	0.08	0.03	0.007	0.006	0.006	0.25	0.84	0.02	0.001
Poland	40.0	0.03	0.08	0.03	0.007	0.006	0.006	0.26	0.87	0.02	0.001
Russian Federation	55.7	0.03	0.12	0.04	0.010	0.009	0.008	0.36	1.21	0.02	0.001

^1^ EDI: estimated daily intake; ^2^ THQ: target hazard quotient.

**Table 5 molecules-26-01229-t005:** The concentration of oxysterols in the fish muscle of selected fish species (mean ± SD) (μg/g).

Oxysterols	Perch	Flounder	Turbot	Plaice	Cod	Bream	Crucian	LOD *
Squalene	14.5 ± 4.7 ^ab^	19.0 ± 1.8 ^ab^	16.0 ± 1.5 ^ab^	23.7 ± 0.80 ^ab^	16.9 ± 1.2 ^ab^	12.2 ± 2.9 ^b^	33 ± 16 ^a^	0.08
Cholesterol	600 ± 130 ^a^	534 ± 78 ^a^	359 ± 32 ^a^	283 ± 39 ^a^	454 ± 37 ^a^	564 ± 40 ^a^	460 ± 140 ^a^	0.06
7-ketocholesterol	1.07 ± 0.32 ^a^	1.44 ± 0.23 ^a^	1.72 ± 0.20 ^a^	0.78 ± 0.14 ^a^	1.79 ± 0.06 ^a^	1.04 ± 0.07 ^a^	1.65 ± 0.81 ^a^	0.02
7α-hydroxycholesterol	0.45 ± 0.14 ^a^	0.68 ± 0.09 ^a^	0.71 ± 0.08 ^a^	0.36 ± 0.05 ^a^	0.77 ± 0.02 ^a^	0.48 ± 0.06 ^a^	0.75 ± 0.41 ^a^	0.01
7β-hydroxycholesterol	1.01 ± 0.44 ^a,b^	1.6 ± 1.1^a,b^	0.50 ± 0.04 ^b^	0.91 ± 0.10 ^a,b^	0.80 ± 0.05 ^a^^,b^	3.2 ± 3.0 ^a^	0.71 ± 0.32 ^a,b^	0.01
5β,6β-epoxycholesterol	0.98 ± 0.23 ^a^	1.27 ± 0.20^a^	1.36 ± 0.15 ^a^	0.66 ± 0.03 ^a^	1.49 ± 0.05 ^a^	1.07 ± 0.07 ^a^	1.53 ± 0.66 ^a^	0.05
5α,6α-epoxycholesterol	0.67 ± 0.21^a^	0.98 ± 0.20 ^a^	1.16 ± 0.15 ^a^	0.71 ± 0.08 ^a^	2.05 ± 0.22 ^a^	0.78 ± 0.27 ^a^	1.45 ± 0.92 ^a^	0.02
Sum of COPs	4.18 ^a^	6.00 ^a^	5.45 ^a^	3.42 ^a^	6.90 ^a^	6.57 ^a^	6.09 ^a^	-
Percent of cholesterol	0.70 ^b^	1.12 ^ab^	1.52 ^a^	1.21 ^ab^	1.52 ^ab^	1.17 ^ab^	1.33 ^ab^	-

^a,b^ homogenous groups in rows (multiple comparisons of mean rank; *p* < 0.050) * LOD—limit of detection (μg/g).

**Table 6 molecules-26-01229-t006:** Instrumental conditions of LC-MS.

Parameter	Value
Curtain gas	35 psi
Ion source gas 1	60 psi
Ion source gas 2	40 psi
Collision gas	“medium”
Ion spray voltage	5000 V
Source temperature	600 °C
Analytical column	Kinetex RP-18 column (100 mm × 4.6 mm, 2.6 µm), Phenomenex (Torrance, CA, USA)
Mobile phase	**A**: HPLC grade water with 0.2% formic acid; **B**: acetonitrile with 0.2% formic acid as eluent B
Flow rate	0.5 mL/min
Gradient (%B)	0 min. 10%; 1 min. 10%; 25 min. 90%, 35 min. 90%
Separation temperature	40 °C
Injection volume	10 µL

**Table 7 molecules-26-01229-t007:** Instrumental conditions of ICP-MS.

Parameter	Value
RF Power	1550 W
Plasma argon flow rate	15 L/min
Auxiliary argon flow rate	0.9 L/min
Nebulizer argon flow rate	1.03 L/min
Cell gas flow rate	4.3 mL/min
Dwell time	0.3 s
Sweeps	100
Number of readings per replicate	3
Conditions	Oxide 156/140 and doubly charged 70/140 <2%

**Table 8 molecules-26-01229-t008:** Instrumental conditions of GC-TOF-MS.

Parameter	Value
Analytical column	Rxi^®^-17SilMS (30 m × 0.25 mm × 0.25 μm, Restek, Bellefonte, Pennsylvania, US)
Carrier gas	helium (purity: 6.0); flow: 1 mL/min
Injector temperature	290 °C
Temperature program	200 °C for 4.60 min; increase 5 °C/min to 290 °C, 290 °C–12.4 min
Transfer line temperature	290 °C
Ion source temperature	250 °C
Ion source energy	70 eV
Injection volume	1 µL
Injection mode	splitless

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
