# Peer review of "Determination of Pharmaceuticals, Heavy Metals, and Oxysterols in Fish Muscle"

_molecules, 2021, doi:10.3390/molecules26051229_

Round 1

Reviewer 1 Report

The topic of the manuscript, determination of pharmaceuticals, toxic elements and oxysterols in fish muscle, is very important and of high interest for the consumer. I think it is also good that six different kind of fish were analyzed. Even though the topic of the study is very important, the quality of the manuscript needs to be improved.  

The introduction is too long and includes mainly general information, which are not relevant for this topic. The topic of the MS is not about fish consumption (guidelines, or amount) it is about how much e.g. pharmaceutical can be detected in fish. I miss a general introduction related to the topic. 2 Tables regarding fish consumption, is too much.

On the other hand, the first paragraph of the discussion section (line 212-235) belongs to the introduction and has nothing to do with discussion. In the discussion section, the results should be discussed, and not how much pharmaceutics are on the market.

Furthermore, I think the order of the MS is confusing, why material and method at the end? It is interesting how many fish were analyzed and how, before we read the results and discussion section.  

My comments in general:

  • Empty space between the numbers or at least between “–“ e.g. line 35: 0.78±14-1.79±0.06 µg/g -> it is very difficult to read, better: 0.78 ± 0.14 – 1.79 ± 0.06 µg/g
  • 2 decimals instead of 3 after the dot, be consistent -> in general use 3 decimals: meaning 123 or 12.3 or 1.23, this is how data should be explained, it is not important if 126,66 ±22, it would be correct: 126 ± 1.22, then the tables are not overloaded with numbers
  • The title of the tables and the headline of the table are sometime the same, skip the headline or change the title of the table.

Abstract:

Line 31: why respectively?

Line 31: better: …fish muscle was between 282 (plaice) and 595 (perch)

Line 34: 6.9 (cod)-> µg/g is missing

Introduction:

Line 48: why is “intake” moved to the right?

Line 58: double check wording: so as to

Line 65: EUMOFA, reference is missing

Line 71: [9-10] should be [9, 10]

Table 2: EU-28: meaning? United States: wrong font size

Results:

Line 105-111 and Table 3: be consistent with the decimals, it should be 2, not 3 decimals

Table 3: Number 150 is written above pharmaceutical

samples were detected pharmaceuticals/total samples-> meaning?  I don’t understand what 2/6 means, 2 out of 6 samples have Metronidazole, and the value is then just from this 2 samples or the average from all 6 samples?

Table 4: meaning of the numbers?

Table 4: Empty space below Amlodipine, why?

2.2 Determination of selected trace elements

Line 123: why respectively

Table 5: Trace elements: abbreviation is missing, I know the abbreviations of the elements, but maybe other readers doesn’t, so either you stick to the full name e.g. mercury instead of Hg or you include the names in the table.

Line 123: the highest level of mercury… if I see it correctly there is just one sample in perch which has such high value, the others are lower, and  similar to the results from flounder, it should be better stated in the text.

Did you compare all the results statistically e.g. tukey post hoc test, how is the p-value?

Line 127-132: does not belong to the results part, it belongs to the discussion part

Figure 1: different order according to the text would be better, starting with cadmium, mercury, arsenic, lead, nickel

x-axis-legend is missing

2.3 Determination of oxysterols

Line 134, 136 and 137: mg/kg is missing

Line 139 and 140: µg/g is missing

3.Discussion

Line 212-235: belongs to introduction

Line 262: J. multidentata: latin name?

Line 265: zebra mussel: latin name?

Line 268-269: check language

Line 270: instead of was should be were

Line 304-318: exact copy of results section, with all the mistakes (comments see above) =-> this is not possible! Should be rewritten!

Line 337: cod: mg/kg is missing; between number empty space “-“

Material and methods

General comments: you cannot have twice Instrumental analyses, then you should call it depending on the analyses: HPLC analyses for the detection of pharmaceutical…for example, furthermore, you separate instrumental analyses for HPLC and GC, what about the determination of trace elements via ICP-MS, should be also separated then as well-> rewrite/reorganize this section

Line 369: how many samples were taken? Is missing in the text.

Line 380: homogenized

Line 383: which centrifuge was used? State this information in the text

Line 411: follow

Line 420: preparation

Line 433: empty space too much between with and an

Line 443: empty space too much between 15 and L

Reference:

Line 520: Services) bracket too much

Author Response

Revision of a Manuscript

Journal:  Molecules

Manuscript ID: molecules-1101085

Title: Determination of pharmaceuticals, toxic elements and oxysterols in fish muscle. Risk assessment for consumers. A pilot study.

Authors: Barbara Bobrowska-Korczak, Agnieszka Stawarska, Arkadiusz Szterk, Karol Ofiara, MaÅ‚gorzata Czerwonka, Joanna GiebuÅ‚towicz.   

Reviewer 1

Thank you very much for your comments, which helped us to improve the quality of our paper.

Reviewer 1.

Comments by Reviewer #1

Changes by Authors

The introduction is too long and includes mainly general information, which are not relevant for this topic. The topic of the MS is not about fish consumption (guidelines, or amount) it is about how much e.g. pharmaceutical can be detected in fish. I miss a general introduction related to the topic. 2 Tables regarding fish consumption, is too much.

Thank you for that remark. The introduction and discussion section have been revised.

On the other hand, the first paragraph of the discussion section (line 212-235) belongs to the introduction and has nothing to do with discussion. In the discussion section, the results should be discussed, and not how much pharmaceutics are on the market.

Thank you for that remark. The introduction and discussion section have been revised.

Furthermore, I think the order of the MS is confusing, why material and method at the end? It is interesting how many fish were analyzed and how, before we read the results and discussion section.  

Thank you for that remark. According to the formatting requirements for Molecules the section “Material and Methods’ should be after “Discussion”.

We added an appropriate information

Empty space between the numbers or at least between “–“ e.g. line 35: 0.78±14-1.79±0.06 µg/g -> it is very difficult to read, better: 0.78 ± 0.14 – 1.79 ± 0.06 µg/g

Thank you for remarks. We would like to apologize for this omission. It was corrected.

2 decimals instead of 3 after the dot, be consistent -> in general use 3 decimals: meaning 123 or 12.3 or 1.23, this is how data should be explained, it is not important if 126,66 ±22, it would be correct: 126 ± 1.22, then the tables are not overloaded with numbers

Thank you for that valuable remark. The results section was revised.

The title of the tables and the headline of the table are sometime the same, skip the headline or change the title of the table,

Thank you for that remark. Taking into account the Reviewer’s suggestion we changed the sentences.

Abstract:

Line 31: why respectively?

Line 31: better: …fish muscle was between 282 (plaice) and 595 (perch)

Line 34: 6.9 (cod)-> µg/g is missing

Thank you for remarks.  Taking into account the Reviewer’s suggestion we changed the sentences.

Introduction:

Line 48: why is “intake” moved to the right?

Line 58: double check wording: so as to

Line 65: EUMOFA, reference is missing

Line 71: [9-10] should be [9, 10]

Table 2: EU-28: meaning? United States: wrong font size

Thank you for remarks.  Taking into account the Reviewer’s suggestion we changed the sentences.

Results:

Line 105-111 and Table 3: be consistent with the decimals, it should be 2, not 3 decimals

Table 3: Number 150 is written above pharmaceutical

samples were detected pharmaceuticals/total samples-> meaning?  I don’t understand what 2/6 means, 2 out of 6 samples have Metronidazole, and the value is then just from this 2 samples or the average from all 6 samples?

Table 4: meaning of the numbers?

Table 4: Empty space below Amlodipine, why?

Thank you for remarks.  Taking into account the Reviewer’s suggestion the results section was revised.

2.2 Determination of selected trace elements

Line 123: why respectively

Table 5: Trace elements: abbreviation is missing, I know the abbreviations of the elements, but maybe other readers doesn’t, so either you stick to the full name e.g. mercury instead of Hg or you include the names in the table.

Line 123: the highest level of mercury… if I see it correctly there is just one sample in perch which has such high value, the others are lower, and  similar to the results from flounder, it should be better stated in the text.

Did you compare all the results statistically e.g. tukey post hoc test, how is the p-value?

Line 127-132: does not belong to the results part, it belongs to the discussion part

Figure 1: different order according to the text would be better, starting with cadmium, mercury, arsenic, lead, nickel

x-axis-legend is missing

Thank you for remarks. We added an appropriate information. The result and discussion part was revised.

2.3 Determination of oxysterols

Line 134, 136 and 137: mg/kg is missing

Line 139 and 140: µg/g is missing

Thank you for remarks. We would like to apologize for this omission. It was corrected.

3.Discussion

Line 212-235: belongs to introduction

Line 262: J. multidentata: latin name?

Line 265: zebra mussel: latin name?

Line 268-269: check language

Line 270: instead of was should be were

Thank you for remarks. The introduction and discussion section have been revised.

Line 304-318: exact copy of results section, with all the mistakes (comments see above) =-> this is not possible! Should be rewritten!

Line 337: cod: mg/kg is missing; between number empty space “-“

Thank you for remarks. The discussion section have been revised.

Material and methods

General comments: you cannot have twice Instrumental analyses, then you should call it depending on the analyses: HPLC analyses for the detection of pharmaceutical…for example, furthermore, you separate instrumental analyses for HPLC and GC, what about the determination of trace elements via ICP-MS, should be also separated then as well-> rewrite/reorganize this section

Line 369: how many samples were taken? Is missing in the text.

Line 380: homogenized

Line 383: which centrifuge was used? State this information in the text

Line 411: follow

Line 420: preparation

Line 433: empty space too much between with and an

Line 443: empty space too much between 15 and L

Thank you for remarks. The material and methods section have been revised. We would like to apologize for omission. All were corrected. We added appropriate information.

Reference:

Line 520: Services) bracket too much

Thank you for that remark. The references section was corrected.

Reviewer 2 Report

I have now completed my review on the manuscript Molecules-1101085, entitled “Determination of pharmaceuticals, toxic elements and oxysterols in fish muscle. Risk assessment for consumers. A pilot study”.

I think this is an interesting manuscript, providing data on both pharmaceuticals, elements and cholesterol in commercial fish species.

General comments

I think that the introduction and results part are good. I think the materials and methods part could be improved by structure the information about the three methods similarly. For instance, for elements the information about the purity and purchase of the compound is at the beginning, while oxysterols and drugs start with sample preparation.

The discussion should be structured better. Some of the toxic effects of the metals could be moved to the introduction. Since Risk assessment for consumers is included in the title, I think you should include this more in the discussion. Also, the conclusion should be more specific towards what you have described in the results and discussion part.

Specific comments

  1. Line 48: there is space before the sentence starts.
  2. Line 66: The definition should be included since these numbers overestimates the intake of aquatic food: food supply is defined as food available for human consumption. At country level, it is calculated as the food remaining for human use after deduction of all non-food utilizations (i.e. food = production + imports + stock withdrawals − exports − industrial use − animal feed – seed – wastage − additions to stock). Wastage includes losses of usable products occurring along distribution chains from farm gate (or port of import) up to the retail level. However, such values do not include consumption-level waste (i.e. retail, restaurant and household waste) and therefore overestimates the average amount of food actually consumed.
  3. Line 78: The transition between the two sentences is not good. Rewrite, maybe include an extra sentence on mercury, where you mention that mercury is known to be high in some fish species?
  4. Line 105: Metronidazole is not allowed to use in food production, I think you should discuss the finding of metronidazole in the discussion.
  5. Line 129: EU Commission Directive: the ML is set with unit (mg/kg wet weight), not mg/kg dry mass.
  6. Line 304-318: It is not necessary to repeat all this from the result part.
  7. Line 374: Include the number of samples for each species.
  8. Line 380: Were the hole fillet homogenized, or just the 750 mg. If the hole filet was not homogenized where on the fillet did you take the sample? The fat content can vary and therefore also the concentration of the compounds analyzed for.
  9. Line 387: The degree sign seems to be underscored
  10. Line 404: Should it be both positive and negative mode?
  11. Line 415: Did the MDL and MQL vary between the species?
  12. Line 430: There is space between 210 º C, and degree is underlined.
  13. Line 432: Is there an extra space between of and 100?
  14. Line 434: You introduce the abbreviation (In) for Internal standard here, but if you want to use the abbreviation you should insert in line 381 where Internal Standard first was introduced, and then use the abbreviation throughout the rest of the manuscript.
  15. Line 458: Write three hundred with numbers
  16. Line 486: I don’t agree that it is little awareness. However, it is difficult to estimate the health risks due to combines exposures. Maybe you should rephrase and rather emphasize the important of gathering more knowledge about this subject.
  17. Unit: Sometimes the unit is written with -1 superscript, while other are written with /. For instance, line 130: mg/kg while line 135 have mg kg-1. This should be harmonized.
  18. Include the measurement uncertainty of the methods
  19. Did you check for any matrix effects?
  20. Was a matrix matched calibration curve used to quantify the analyses?
  21. Significant figures: I suggest that you reduce the number of digits that you include in the results. To include 5 digits in the results is too much, the last digits give no meaning to the results.

Tables

  1. I miss information about the tables. For instance, table 4, is it LOD that are included? Table 5, is it mean ±SD?
  2. Table 3: Include the n for Bream and Crucian.
  3. In the tables you use LOD, however the method and material section you use MQL, and LOD is not mention.

Figures

I fail to see an explanation for figure 1. What are the dots? What does the line and box show? Are all analyzed samples included here? It is difficult to see the text on the x-axis and y-axis, there is enough space to increase the text. You have included the isotope number in the figure. If you want to include it here, you should also include it in the text in the method and material section.

Author Response

Revision of a Manuscript

Journal:  Molecules

Manuscript ID: molecules-1101085

Title: Determination of pharmaceuticals, toxic elements and oxysterols in fish muscle. Risk assessment for consumers. A pilot study.

Authors: Barbara Bobrowska-Korczak, Agnieszka Stawarska, Arkadiusz Szterk, Karol Ofiara, MaÅ‚gorzata Czerwonka, Joanna GiebuÅ‚towicz.   

Reviewer 2

Thank you very much for your comments, which helped us to improve the quality of our paper.

Comments by Reviewer #2

Changes by Authors

I think that the introduction and results part are good. I think the materials and methods part could be improved by structure the information about the three methods similarly. For instance, for elements the information about the purity and purchase of the compound is at the beginning, while oxysterols and drugs start with sample preparation.

Thank you for that remark. The materials and methods section have been revised.

The discussion should be structured better. Some of the toxic effects of the metals could be moved to the introduction. Since Risk assessment for consumers is included in the title, I think you should include this more in the discussion. Also, the conclusion should be more specific towards what you have described in the results and discussion part.

Thank you for that remark. The discussion and conclusion section have been revised.

Specific comments

Line 48: there is space before the sentence starts.

Thank you for remarks. We would like to apologize for this omission. It was corrected.

Line 66: The definition should be included since these numbers overestimates the intake of aquatic food: food supply is defined as food available for human consumption. At country level, it is calculated as the food remaining for human use after deduction of all non-food utilizations (i.e. food = production + imports + stock withdrawals − exports − industrial use − animal feed – seed – wastage − additions to stock). Wastage includes losses of usable products occurring along distribution chains from farm gate (or port of import) up to the retail level. However, such values do not include consumption-level waste (i.e. retail, restaurant and household waste) and therefore overestimates the average amount of food actually consumed.

Thank you for that remark. The introduction section have been revised.

Line 78: The transition between the two sentences is not good. Rewrite, maybe include an extra sentence on mercury, where you mention that mercury is known to be high in some fish species?

Thank you for that remark. The introduction section have been revised.

Line 105: Metronidazole is not allowed to use in food production, I think you should discuss the finding of metronidazole in the discussion.

Thank you for that remark. Metronidazole could be use as a drug for people (in Poland) that is why we studied its content.

Line 129: EU Commission Directive: the ML is set with unit (mg/kg wet weight), not mg/kg dry mass.

Thank you for remarks. We would like to apologize for this mistake. It was corrected.

Line 304-318: It is not necessary to repeat all this from the result part

Thank you for that remark. The results and discussion section were revised.

Line 374: Include the number of samples for each species.

Thank you for that remark. We added appropriate information.

Line 380: Were the hole fillet homogenized, or just the 750 mg. If the hole filet was not homogenized where on the fillet did you take the sample? The fat content can vary and therefore also the concentration of the compounds analyzed for

Thank you. We homogenized only the muscle isolated from the same part of the fish each time.

Line 387: The degree sign seems to be underscored

Thank you. We made the correction.

Line 404: Should it be both positive and negative mode?

We made the measurements in positive mode. We corrected the sentence

Line 415: Did the MDL and MQL vary between the species?

We used the muscles only. Thus, we performed the validation assuming the muscles of all fish species as one matrix.

Line 430: There is space between 210 º C, and degree is underlined

Thank you for that remark. We made the correction.

Line 432: Is there an extra space between of and 100?

Thank you. We made the correction.

Line 434: You introduce the abbreviation (In) for Internal standard here, but if you want to use the abbreviation you should insert in line 381 where Internal Standard first was introduced, and then use the abbreviation throughout the rest of the manuscript.

Thank you. We made the correction.

Line 458: Write three hundred with numbers

Thank you. We made the correction.

Line 486: I don’t agree that it is little awareness. However, it is difficult to estimate the health risks due to combines exposures. Maybe you should rephrase and rather emphasize the important of gathering more knowledge about this subject.

Thank you for that remark. The conclusion section was revised.

Unit: Sometimes the unit is written with -1 superscript, while other are written with /. For instance, line 130: mg/kg while line 135 have mg kg-1. This should be harmonized.

Thank you for that remark. We made the correction.

Include the measurement uncertainty of the methods

The appropriate sentence was added to the manuscript.

Did you check for any matrix effects?

Yes, in some cases the matrix effect was significant, thus we used a matrix matched calibration curve for quantitation.

Was a matrix matched calibration curve used to quantify the analyses?

Yes,  the matrix matched calibration was used to quantify the analyses. The appropriate data was added to the text.

Significant figures: I suggest that you reduce the number of digits that you include in the results. To include 5 digits in the results is too much, the last digits give no meaning to the results.

Thank you for that remark. We made the correction.

I miss information about the tables. For instance, table 4, is it LOD that are included? Table 5, is it mean ±SD?

The Table was created according to Reviewer suggestion.

Table 3: Include the n for Bream and Crucian

Thank you for that remark. We added appropriate information.

In the tables you use LOD, however the method and material section you use MQL, and LOD is not mention.

Thank you for that remark. The material and methods and results section was revised.

Figures

I fail to see an explanation for figure 1. What are the dots? What does the line and box show? Are all analyzed samples included here? It is difficult to see the text on the x-axis and y-axis, there is enough space to increase the text. You have included the isotope number in the figure. If you want to include it here, you should also include it in the text in the method and material section.

Thank you for that remark. The Figure was omitted.

Reviewer 3 Report

ManuscriptID: molecules-1101085 - Determination of pharmaceuticals, toxic elements and oxysterols in fish muscle. Risk assessment for consumers. A pilot study.

The manuscript is very interesting and an important topic for public health, environmental and food science nowadays, however there are some severe weakness.

As stated by authors, the aim of the study was to assess the levels of 115 multi-class pharmaceuticals and other substances, but the title is misleading.

In the introduction the topic it’s not clear; for example no mention about potential damages derived from oxysterols, compounds that were then researched.

The results are interesting but don’t support  the discussions.

  • the sample is too small for a risk assessment;  No statistical analysis was performed; no mention on the choice of the selected fish species ( are these species particularly sold in the Baltic regions?); most of Discussion is not a discussion but results yet mentioned and state of art on the “toxics”  topic of the work

English must be improved

For this reason I suggest to do reject the paper

Specific suggestions

  • Title: the authors have to be more specific: toxic elements could be switched into heavy metals and trace elements,
  • LINE 45: .. source of proteins… please add references
  • LINE 48-49 :… Fish protein… how a protein could be a source of lipids?
  • LINE 50-51:… Diet rich in fish protein lowers the risk of cardiovascular… it’s for lipids or for protein? please add references
  • LINE 56:… please add more recent references; in addition, have you considered literature about methyl-mercury ingestion through big pelagic fish (as tuna)?
  • Line 65: there is a more recent EUMOFA panel, please use it as reference
  • LINE 77: please add references
  • LINE 82: I think that other potential hazards are parasites.. I think you have to mention also this hazard (lots of literature is present on this topic)
  • Line 100-101: it’s a personal consideration, not a result
  • Section 2.1: in order to make reading more simple, please put the results only into table
  • Section 2.2: mercury, cadmium and lead are not trace elements, but heavy metals
  • Section 2.2: also for this section, in order to make reading more simple, please put the results only into table
  • Section 2.3: also for this section, in order to make reading more simple, please put the results only into table
  • Table n. 5: the average level of trace element is followed by standard error or standard deviation? Please clarify
  • Figure n. 1: please explain the plots into the figures
  • Table n.5: the average level of oxysterols is followed by standard error or standard deviation? Please clarify
  • Line 212: please add references
  • Line 219: please support the sentences with a reference
  • Line 212-233: I believe that it’s not a discussion, but a state of art, potentially present in the introduction section
  • Line 236-240: it’s a result (yet mentioned in the previous section)
  • Line 240-270: there is no discussion about the study result and the literature, but only a mention of studies
  • Line 273: please add references
  • Line 304-312: the sentences repeat the result yet mentioned into the previous section
  • Line 318: all the references are not right: the EU regulation that have to be mentioned in REG UE 1881/06 and subsequent modification
  • Line 319-334: it’s not a discussion, but an introduction on the oxysterols
  • Line 383 and within the text: 10000 “g”? what means? Grams? Or RPM
  • Section 4.3 – could you insert all instrumental parameters into a table?
  • Line 423: approximately: in an international paper is not possible to approximate a sample weight, especially if it’s a little sample

the topic is interesting but there are many lacks in english and methods and in my opinion conclusions  are not supported by results

Author Response

Revision of a Manuscript

Journal:  Molecules

Manuscript ID: molecules-1101085

Title: Determination of pharmaceuticals, toxic elements and oxysterols in fish muscle. Risk assessment for consumers. A pilot study.

Authors: Barbara Bobrowska-Korczak, Agnieszka Stawarska, Arkadiusz Szterk, Karol Ofiara, MaÅ‚gorzata Czerwonka, Joanna GiebuÅ‚towicz.   

Reviewer 3.

Thank you very much for your comments, which helped us to improve the quality of our paper.

Comments by Reviewer #3

Changes by Authors

As stated by authors, the aim of the study was to assess the levels of 115 multi-class pharmaceuticals and other substances, but the title is misleading.

Thank you for that remark. The title was revised.

In the introduction the topic it’s not clear; for example no mention about potential damages derived from oxysterols, compounds that were then researched.

Thank you for that remark. The introduction section have been revised.

The results are interesting but don’t support  the discussions.

  • the sample is too small for a risk assessment;  No statistical analysis was performed; no mention on the choice of the selected fish species ( are these species particularly sold in the Baltic regions?); most of Discussion is not a discussion but results yet mentioned and state of art on the “toxics”  topic of the work

Thank you for that remark. The introduction and discussion section have been revised.

English must be improved

English was checked by native speaker.

Title: the authors have to be more specific: toxic elements could be switched into heavy metals and trace elements,

Thank you for that remark. The title was revised.

LINE 45: .. source of proteins… please add references

Thank you for that remark. We added appropriate information.

LINE 48-49 :… Fish protein… how a protein could be a source of lipids?

LINE 50-51:… Diet rich in fish protein lowers the risk of cardiovascular… it’s for lipids or for protein? please add references

Thank you for that remark. The introduction section have been revised. We added appropriate information.

LINE 56:… please add more recent references; in addition, have you considered literature about methyl-mercury ingestion through big pelagic fish (as tuna)?

Thank you for that remark. We added appropriate information.

Line 65: there is a more recent EUMOFA panel, please use it as reference

Thank you for that remark. The introduction section have been revised.

LINE 77: please add references

Thank you for that remark. We added appropriate information.

LINE 82: I think that other potential hazards are parasites.. I think you have to mention also this hazard (lots of literature is present on this topic)

Thank you for that remark. We added appropriate information.

Line 100-101: it’s a personal consideration, not a result

Thank you. We made the correction.

Section 2.1: in order to make reading more simple, please put the results only into table

Thank you for that remark. The description of the results has been shortened as much as possible.

Section 2.2: mercury, cadmium and lead are not trace elements, but heavy metals

Thank you for that remark. The results section have been revised.

Section 2.2: also for this section, in order to make reading more simple, please put the results only into table

Thank you for that remark. The description of the results has been shortened as much as possible.

Section 2.3: also for this section, in order to make reading more simple, please put the results only into table

Thank you for that remark. The description of the results has been shortened as much as possible.

Table n. 5: the average level of trace element is followed by standard error or standard deviation? Please clarify

Thank you for that remark. We added appropriate information.

Figure n. 1: please explain the plots into the figures

Thank you for that remark. The Figure was omitted.

Table n.5: the average level of oxysterols is followed by standard error or standard deviation? Please clarify

Thank you for that remark. We added appropriate information.

Line 212: please add references

Thank you for that remark. We added appropriate information.

Line 219: please support the sentences with a reference

Thank you for that remark. We added appropriate information.

Line 212-233: I believe that it’s not a discussion, but a state of art, potentially present in the introduction section

Thank you for that remark. The introduction and discussion section have been revised.

Line 236-240: it’s a result (yet mentioned in the previous section)

Thank you for that remark. The discussion section have been revised.

Line 240-270: there is no discussion about the study result and the literature, but only a mention of studies

Thank you for that remark. We added appropriate information.

Line 273: please add references

Thank you for that remark. We added appropriate information.

Line 304-312: the sentences repeat the result yet mentioned into the previous section

Thank you. We made the correction.

Line 318: all the references are not right: the EU regulation that have to be mentioned in REG UE 1881/06 and subsequent modification

Thank you. All presented regulation are amended Regulation (EC) No 1881/2006.

Line 319-334: it’s not a discussion, but an introduction on the oxysterols

Thank you. We made the correction.

Line 383 and within the text: 10000 “g”? what means? Grams? Or RPM

Thank you. We made the correction.

Line 423: approximately: in an international paper is not possible to approximate a sample weight, especially if it’s a little sample

Thank you. We made the correction.

the topic is interesting but there are many lacks in english and methods and in my opinion conclusions  are not supported by results

Thank you. We made the correction.

Round 2

Reviewer 1 Report

The manuscript has improved, especially the introduction section.

I still have some general comments:

  • The page numbers are not correct
  • For consistence, please check again that you use just 3 numbers. As example: abstract line 33 (283±39 mg/kg-> should be 283±9 mg/kg) or under section 2.2: 68±26 or 12.8±2.3 or 11.19±0.93…
  • You included Latin names for fish species, but didn’t write the general name, like cod (Gadus morhua callarias), please check e.g. line 378
  • The discussion section should include just the discussion of the results, it shouldn’t include calculations or formulas, those parts belong to material/methods and results section (especially the tables)
  • Please have another critical look for the empty space between the numbers or especially between “-“ (e.g. line 466)

Introduction:

Line 48: is instead of are

Line 121: arsenic-induced without “-“, two words

Line 142: please reformulate: that is why…(expression) -> The present study…

Line 148: Numbers till 12 should be written out

Results:

Table 1: I still don’t get the point with n=2/6. It means that 2 samples were positive and 6 samples were analyzed. The numbers which are stated is then just from this 2 samples or the average from all 6 samples?

LOD Sulfadimethoxine should be 0.10

LOD in the table should be marked with * and the underneath the table * LOD -limit…, so you know this belongs together

Table 2: LOD abbreviation underneath the table is missing

Title: Pharmaceuticals and selected metabolites which were not detected…

2.2 Determination of selected trace elements

Be consistent with the decimals, it should be 3 and not 2, check text and table 3

Figure 1 is not mentioned in the text, is it included?

Figure 1: Figure text is missing as well as x-axis-legend

2.3 Determination of oxysterols

You use mg/kg as unit in the text, but in table 4 you call it µg/g -> be consistent

Line 197: check numbers

Discussion

Line 278: of 11 out 98

Line 295, 308 and 328, 378: fish name in front of latin name is missing

Line 314: from not form

Line 315: numbers till 12 should be written out

Line 331: latin name in brackets

Line 348: that is why, sounds not scientific: therefore, …

Line 370: close to-> similar

Line 403-446: formula belongs to material/method section and not to discussion

Table 6: result section, not to discussion

Line 415-420: those are results, they should be in the results section, pick some out and discuss it

Line 457: check intake consumption, decide for one word, both means the same

Line 460: rewrite: the reason is that since this…not scientific

Line 462: this is as reported by others…what do you want to write, I don’t get this sentence

Line 464: without respectively

Line 466: include empty space between “-“

Material/methods

Line 526: afterwards

Line 555: one

Line 609: due to variables in any group…what do you mean with any group?

Line 622: considered as a serious problem

Author Response

Reviewer 1.

Revision of a Manuscript

Journal:  Molecules

Manuscript ID: molecules-1101085

Title: Determination of pharmaceuticals, toxic elements and oxysterols in fish muscle. Risk assessment for consumers. A pilot study.

Authors: Barbara Bobrowska-Korczak, Agnieszka Stawarska, Arkadiusz Szterk, Karol Ofiara, Małgorzata Czerwonka, Joanna Giebułtowicz

Thank you very much for your comments, which helped us to improve the quality of our paper.

Comments by Reviewer #1

Changes by Authors

The page numbers are not correct

Thank you for that remark. We made the correction.

For consistence, please check again that you use just 3 numbers. As example: abstract line 33 (283±39 mg/kg-> should be 283±9 mg/kg) or under section 2.2: 68±26 or 12.8±2.3 or 11.19±0.93.

2.2 Determination of selected trace elements

Be consistent with the decimals, it should be 3 and not 2, check text and table 3

Thank you for the remark on too high accuracy of presented results. We explored the subject and asked for help our colleagues from certified laboratory. According to guideline “Good Laboratory Practice for Rounding Expanded Uncertainties and Calibration Values” the uncertainty should be round off to two significant digits. The mean should be round off to the same number of decimal places as uncertainty. As a results depending on the size of the standard deviation, the mean can have different number of decimal places even for the same variable.    

You included Latin names for fish species, but didn’t write the general name, like cod (Gadus morhua callarias), please check e.g. line 378

Thank you for that remark. We made the correction.

The discussion section should include just the discussion of the results, it shouldn’t include calculations or formulas, those parts belong to material/methods and results section (especially the tables)

Thank you for that remark. The discussion, results and material/methods section have been revised. The formulas was moved to the results.

Please have another critical look for the empty space between the numbers or especially between “-“ (e.g. line 466)

Thank you for that remark. We made the corrections.

Introduction:

Line 48: is instead of are

Line 121: arsenic-induced without “-“, two words

Line 142: please reformulate: that is why…(expression) -> The present study…

Line 148: Numbers till 12 should be written out

Thank you for that remark. We made the correction.

Table 1: I still don’t get the point with n=2/6. It means that 2 samples were positive and 6 samples were analyzed. The numbers which are stated is then just from this 2 samples or the average from all 6 samples?

We corrected the description to be more readable. The table title was changed to: The maximum concentration of pharmaceuticals detected in fish muscles (ng/g) and the number of positive findings (np)   

LOD Sulfadimethoxine should be 0.10

LOD in the table should be marked with * and the underneath the table * LOD -limit…, so you know this belongs together

Table 2: LOD abbreviation underneath the table is missing

Title: Pharmaceuticals and selected metabolites which were not detected…

Thank you for that remark. We made the correction. We explained the abbreviations underneath all tables.

Figure 1 is not mentioned in the text, is it included?

Figure 1: Figure text is missing as well as x-axis-legend

Thank you for that remark. The Figure has been deleted.

2.3 Determination of oxysterols

You use mg/kg as unit in the text, but in table 4 you call it µg/g -> be consistent

Line 197: check numbers

Thank you for remarks. We would like to apologize for this omission. It was corrected.

Discussion

Line 278: of 11 out 98

Line 295, 308 and 328, 378: fish name in front of latin name is missing

Line 314: from not form

Line 315: numbers till 12 should be written out

Line 331: latin name in brackets

Line 348: that is why, sounds not scientific: therefore, …

Line 370: close to-> similar

Line 403-446: formula belongs to material/method section and not to discussion

Table 6: result section, not to discussion

Line 415-420: those are results, they should be in the results section, pick some out and discuss it

Line 457: check intake consumption, decide for one word, both means the same

Line 460: rewrite: the reason is that since this…not scientific

Line 462: this is as reported by others…what do you want to write, I don’t get this sentence

Line 464: without respectively

Line 466: include empty space between “-“

Thank you for that remark. We made the correction.

Material/methods

Line 526: afterwards

Line 555: one

Line 609: due to variables in any group…what do you mean with any group?

Line 622: considered as a serious problem

Thank you for remarks. The material and methods section have been revised accordingly.

Line 609” Due to lack of normal distribution of variables in groups, as assessed by Shapiro–Wilk test…”

Reviewer 3 Report

The suggested reviews were made, but it’s difficult to follow the revised version of the manuscript because of the many revisions (as suggested by the n. 3 revisors).

The introduction (too long) was revised but many space was dedicated to the benefits of fish consumption leaving out the topic of the research. It seems that there is no link between the different “sections” of the introduction (benefits, risk, pharmaceuticals, metals etc). The manuscript needs improvements in section division (often results are found in discussions, methods are found in the introduction and introduction in discussions). For example at Line 388-450: it’ not a discussion, but it’s a simple comment of a risk assessment  “scenario” in different nations. Please delete this section or put into the introduction shortening the section

Specific suggestions:

Line 80: please cite

  • Smaldone, Giorgio, Abollo, Elvira, Marrone, Raffaele, Bernardi, Cristian E. M., Chirollo, Claudia, Anastasio, Aniello, del Hierro, Santiago P. (2020). Risk-based scoring and genetic identification for anisakids in frozen fish products from Atlantic FAO areas.
  • Ariano A., Marrone R., Andreini R., Smaldone G., Velotto S., Montagnaro S., Anastasio A., Severino L. (2019). Metal concentration in muscle and digestive gland of common octopus (Octopus vulgaris) from two coastal site in Southern Tyrrhenian Sea (Italy). MOLECULES, vol. 24, ISSN: 1420-3049, doi: 10.3390/molecules24132401

Line 105-125 The “metal section” is too long, I suggest shortening this part.

Line 149-152: It’s a method, please insert in the methods section

Sections 2.1, 2.2, 2.3:  as stated in the previous round of revision, to make reading simpler, please put the results only into tables

Figures: please add the mean value of the selected metal, not only the range

Line 273-276: it’s a method yet mentioned

Line 300: please cite: Smaldone G, Marrone R, Cappiello S, Martin G A, Oliva G, Cortesi M L, Anastasio A (2014). Occurrence of antibiotic resistance in bacteria isolated from seawater organisms caught in Campania Region: preliminary study. BMC VETERINARY RESEARCH, vol. 10, ISSN: 1746-6148,

Line 357-364: it’s a result, please put into results section

Line 464-467- it’s a result, please put into res

Author Response

Reviewer 3.

Revision of a Manuscript

Journal:  Molecules

Manuscript ID: molecules-1101085

Title: Determination of pharmaceuticals, toxic elements and oxysterols in fish muscle. Risk assessment for consumers. A pilot study.

Authors: Barbara Bobrowska-Korczak, Agnieszka Stawarska, Arkadiusz Szterk, Karol Ofiara, MaÅ‚gorzata Czerwonka, Joanna GiebuÅ‚towicz     

Thank you very much for your comments, which helped us to improve the quality of our paper.

Comments by Reviewer #3

Changes by Authors

The introduction (too long) was revised but many space was dedicated to the benefits of fish consumption leaving out the topic of the research. It seems that there is no link between the different “sections” of the introduction (benefits, risk, pharmaceuticals, metals etc). The manuscript needs improvements in section division (often results are found in discussions, methods are found in the introduction and introduction in discussions). For example at Line 388-450: it’ not a discussion, but it’s a simple comment of a risk assessment  “scenario” in different nations. Please delete this section or put into the introduction shortening the section.

Thank you for remarks. The introduction section have been abbreviated.

The discussion, results and material/methods section have been revised.

Specific suggestions:

Line 80: please cite

  • Smaldone, Giorgio, Abollo, Elvira, Marrone, Raffaele, Bernardi, Cristian E. M., Chirollo, Claudia, Anastasio, Aniello, del Hierro, Santiago P. (2020). Risk-based scoring and genetic identification for anisakids in frozen fish products from Atlantic FAO areas.
  • Ariano A., Marrone R., Andreini R., Smaldone G., Velotto S., Montagnaro S., Anastasio A., Severino L. (2019). Metal concentration in muscle and digestive gland of common octopus (Octopus vulgaris) from two coastal site in Southern Tyrrhenian Sea (Italy). MOLECULES, vol. 24, ISSN: 1420-3049, doi: 10.3390/molecules24132401

Thank you. The references have been added.

Line 105-125 The “metal section” is too long, I suggest shortening this part.

Thank you for remarks. The “metal section” section have been abbreviated.

Line 149-152: It’s a method, please insert in the methods section

Thank you for that remark. We made the correction.

Sections 2.1, 2.2, 2.3:  as stated in the previous round of revision, to make reading simpler, please put the results only into tables

Thank you for that remark. We corrected the results section to be more readable.

Figures: please add the mean value of the selected metal, not only the range

Thank you for that remark. The Figure has been deleted.

Line 273-276: it’s a method yet mentioned

Thank you for that remark. We made the correction.

Line 300: please cite: Smaldone G, Marrone R, Cappiello S, Martin G A, Oliva G, Cortesi M L, Anastasio A (2014). Occurrence of antibiotic resistance in bacteria isolated from seawater organisms caught in Campania Region: preliminary study. BMC VETERINARY RESEARCH, vol. 10, ISSN: 1746-6148,

Thank you. The references have been cited.

Line 357-364: it’s a result, please put into results section

Thank you for that remark. We made the correction.
